# The Performance of a Calcaneal Quantitative Ultrasound Device, CM-200, in Stratifying Osteoporosis Risk among Malaysian Population Aged 40 Years and Above

**DOI:** 10.3390/diagnostics10040178

**Published:** 2020-03-25

**Authors:** Shaanthana Subramaniam, Chin-Yi Chan, Ima Nirwana Soelaiman, Norazlina Mohamed, Norliza Muhammad, Fairus Ahmad, Pei-Yuen Ng, Nor Aini Jamil, Noorazah Abd Aziz, Kok-Yong Chin

**Affiliations:** 1Department of Pharmacology, Faculty of Medicine, Universiti Kebangsaan Malaysia, Jalan Yaacob Latif, Bandar Tun Razak, Cheras 56000, Malaysia; shaanthana_bks@hotmail.com (S.S.); chanchinyi94@gmail.com (C.-Y.C.); imasoel@ppukm.ukm.edu.my (I.N.S.); azlina@ppukm.ukm.edu.my (N.M.); norliza_ssp@ppukm.ukm.edu.my (N.M.); 2Department of Anatomy, Faculty of Medicine, Universiti Kebangsaan Malaysia, Jalan Yaacob Latif, Bandar Tun Razak, Cheras 56000, Malaysia; apai.kie@gmail.com; 3Drug and Herbal Research Centre, Faculty of Pharmacy, Universiti Kebangsaan Malaysia Kuala Lumpur Campus, Jalan Raja Muda Abdul Aziz, Kuala Lumpur 50300, Malaysia; pyng@ukm.edu.my; 4Centre for Community Health Studies, Faculty of Health Science, Universiti Kebangsaan Malaysia Kuala Lumpur Campus Jalan Raja Muda Abdul Aziz, Kuala Lumpur 50300, Malaysia; ainijamil@ukm.edu.my; 5Department of Family Medicine, Faculty of Medicine, Universiti Kebangsaan Malaysia, Jalan Yaacob Latif, Bandar Tun Razak, Cheras 56000, Malaysia; azah@ppukm.ukm.edu.my

**Keywords:** bone, dual-energy X-ray absorptiometry, quantitative ultrasound, sensitivity, specificity

## Abstract

Background: Calcaneal quantitative ultrasound (QUS) is widely used in osteoporosis screening, but the cut-off values for risk stratification remain unclear. This study validates the performance of a calcaneal QUS device (CM-200) using dual-energy X-ray absorptiometry (DXA) as the reference and establishes a new set of cut-off values for CM-200 in identifying subjects with osteoporosis. Methods: The bone health status of Malaysians aged ≥40 years was assessed using CM-200 and DXA. Sensitivity, specificity, area under the curve (AUC) and the optimal cut-off values for risk stratification of CM-200 were determined using receiver operating characteristic (ROC) curves and Youden’s index (J). *Results*: From the data of 786 subjects, CM-200 (QUS T-score <−1) showed a sensitivity of 82.1% (95% CI: 77.9–85.7%), specificity of 51.5% (95% CI: 46.5–56.6%) and AUC of 0.668 (95% CI: 0.630–0.706) in identifying subjects with suboptimal bone health (DXA T-score <−1) (*p* < 0.001). At QUS T-score ≤−2.5, CM-200 was ineffective in identifying subjects with osteoporosis (DXA T-score ≤−2.5) (sensitivity 14.4% (95% CI: 8.1–23.0%); specificity 96.1% (95% CI: 94.4–97.4%); AUC 0.553 (95% CI: 0.488–0.617); *p* > 0.05). Modified cut-off values for the QUS T-score improved the performance of CM-200 in identifying subjects with osteopenia (sensitivity 67.7% (95% CI: 62.8–72.3%); specificity 72.8% (95% CI: 68.1–77.2%); J = 0.405; AUC 0.702 (95% CI: 0.666–0.739); *p* < 0.001) and osteoporosis (sensitivity 79.4% (95% CI: 70.0–86.9%); specificity 61.8% (95% CI: 58.1–65.5%); J = 0.412; AUC 0.706 (95% CI: 0.654–0.758); *p* < 0.001). Conclusion: The modified cut-off values significantly improved the performance of CM-200 in identifying individuals with osteoporosis. Since these values are device-specific, optimization is necessary for accurate detection of individuals at risk for osteoporosis using QUS.

## 1. Introduction

The rapid ageing of the global population brings forth many noncommunicable diseases, including osteoporosis and its associated fragility fractures. Apart from bearing the direct and direct medical costs, fractured patients suffer from chronic pain, loss of independence and increased mortality [1]. A pooled analysis of data from the United States, Australia, Asia and the Middle East Crescent shows that hip fractures are particularly devastating because 20–40% of people suffering from hip fractures would die within a year and 10% of the survivors would suffer contralateral hip fractures [2]. By 2050, 50% of all hip fractures worldwide are expected to occur in Asia [3]. Malaysia has been predicted to experience a 3.55-fold increase in hip fracture incidence by 2050 compared to 2018, which is one of the highest increases among Asian countries [4]. Early detection of bone loss and timely prophylaxis are critical in retarding the progression of osteoporosis among high-risk individuals.

Osteoporosis is diagnosed based on bone mineral density (BMD) at the lumbar spine or proximal femur measured by dual-energy X-ray absorptiometry (DXA), which serves as the gold standard technique in clinical settings [5,6]. The individual BMD is then compared against the reference young adult mean values to generate T-score. According to guidelines by World Health Organization (WHO), a BMD value ≤−2.5 standard deviations (SD) of the young adult mean (or a T-score ≤−2.5) indicates osteoporosis, while a T-score between −1 and −2.5 indicates osteopenia [7]. DXA can predict fracture risk and monitor response to treatment [8,9,10]. Moreover, it can determine the whole body and regional body composition [11,12]. Despite its crucial role in the clinical settings, the accessibility of DXA is limited by the cost of the machine and procedure and availability in developing countries. As of 2012, there are about 100 DXA machines available throughout the whole of Malaysia [13]. Hence, it is not suitable to be used for public screening of osteoporosis [13].

Quantitative ultrasound (QUS) offers a portable, radiation-free and relatively less costly method of bone health screening compared to DXA [14]. In general, QUS uses transmission time of ultrasound (speed of sound; SOS) or attenuation of sound signals/frequency (broadband attenuation ultrasound; BUA) across the body part measured to determine bone health [15]. However, the algorithm for generating QUS indices differ between manufacturers and the results may not be comparable. Previous studies demonstrated that QUS indices associate significantly with bone mass, microarchitecture and fracture risk in population studies [16,17]. According to the International Society for Clinical Densitometry (ISCD), the calcaneus is the only validated anatomical site recommended for osteoporosis screening [18]. Previous studies have proved that calcaneal QUS is useful for early detection of osteoporosis [19,20,21]. However, the use of WHO’s T-score cut-offs for osteoporosis and osteopenia in QUS remains controversial because the cut-off values of <−1 and ≤−2.5 are established based on DXA and the skeletal properties examined are different between QUS and DXA [18].

The use of CM-200, a calcaneal QUS device, in population studies to investigate bone health has been reported [15,22,23,24,25,26]. Since this device has built-in reference data for the Japanese population, it has been used to assess the bone health status of the Asian population. Most studies using CM-200 adopted the WHO’s T-score cut-off based on DXA, which is not recommended by ISCD [18]. Thus, the current study aimed to validate the performance of CM-200 against DXA and establish a new set of cut-off values in identifying subjects with osteoporosis in a population aged ≥40 years and residing in Klang Valley, Malaysia.

## 2. Materials and Methods

This cross-sectional study was conducted from April 2018 to April 2019 in Klang Valley, Malaysia. The protocol of the main study has been described previously [27,28,29]. Community-living Malaysian subjects aged ≥40 years were recruited using quota sampling technique, whereby subjects were stratified based on the demographic population in Klang Valley, which is 45% Malays, 45% Chinese and 10% Indians and other ethnic groups [30]. Invitations with specific inclusion and exclusion criteria were sent to community centers in Klang Valley and advertised in local newspapers and radio. Subjects with conditions that alter bone metabolism (hypo/hyperthyroidism, hypo/hypergonadism, hypo/hypercalcemia), previously diagnosed with bone diseases (osteogenesis imperfecta, osteomalacia, Paget’s disease), receiving therapeutic agents (thiazide diuretics, glucocorticoids, osteoporosis treatment agents), having mobility issues, with implants (at lumbar spine, hip and lower limbs) and those who failed to complete the screening procedure were excluded from the study. Informed consent was obtained from all the subjects before their enrolment. Research Ethics Committee of Universiti Kebangsaan Malaysia (Code: UKM PPI/111/8/JEP-2017-761) reviewed and approved the protocol of the main study on 27 December 2017.

Sample size calculation was performed using MedCalc (MedCalc Software Ltd., Ostend, Belgium) to ensure that the number of subjects recruited was sufficient for receiver operating characteristic (ROC) analysis [31]. The following values were used for the calculation: type-1 error = 0.05; type-2 error = 0.20; target area under the curve (AUC) = 0.7; AUC null hypothesis = 0.5; ratio of negative/positive cases = 689/97 (nonosteoporosis/osteoporosis) [27]. The minimal sample size derived was 19 positive cases and 135 negative cases. Thus, the number of subjects recruited for the main study was sufficient for ROC analysis.

All subjects completed a demographic questionnaire prior to their bone health assessment. Their age was determined from records on their identification cards. Their sex, ethnicity and presence of medical conditions were self-declared. Standing height of the subjects without shoes was measured to the nearest 1 cm using a stadiometer (Seca, Hamburg, Germany). Body weight of the subjects with light clothing but without shoes was measured to the nearest 0.1 kg using a weighing scale (Tanita, Tokyo, Japan). Body mass index (BMI: kg/m^2^) was calculated as per the convention. For subjects aged <65 years, BMI was classified as underweight (<18.5 kg/m^2^), normal (18.5–24.9 kg/m^2^) or overweight (>24.9 kg/m^2^) [32]. For subjects aged >65 years, a BMI <22 kg/m^2^ was classified as underweight, 22–27 kg/m^2^ as normal and >27 kg/m^2^ as overweight [33]. For the analysis, both underweight and normal BMI were grouped, while overweight was set as another group. 

BMD at the lumbar spine (anteroposterior, L1-L4) and left hip of the subjects was assessed using DXA (Hologic Discovery QDR Wi densitometer, Hologic, MA, USA) operated by a single trained technician, blinded to the results of QUS, throughout the study period. WHO criteria were referred to categorize the bone health status of subjects as osteoporosis (T-score ≤ −2.5), osteopenia (T-score < −1 and > −2.5) or normal (T-score > −1.0) based on either lumbar spine or left hip T-score values. The ethnic-based reference BMD of Singaporean young adults was used in the calculation of T-score as per the recommendation of Malaysian Osteoporosis Society [34]. This reference was included in the DXA software provided by the manufacturer. The DXA measurements were performed as per the standard protocol. The subjects were required to wear light clothing and lie supine on the DXA machine. The technician positioned the subjects accordingly for the scans. Daily calibration was conducted using a phantom. Short-term in-vivo coefficients of variation for this device were 1.8% and 1.2% for the lumbar spine and total hip, respectively.

Quantitative ultrasound measurement was performed using the gel-based CM-200 bone ultrasonometer (Furuno, Nishinomiya City, Japan) operated by another trained technician, blinded to the results of DXA, throughout the study period. This device uses SOS in m/s as the parameter to assess the bone health status of the subjects. For the present study, T-score generated based on the SOS was used to classify the bone health status of the subjects. T-score values obtained were based on the Japanese population reference provided by the manufacturer. A T-score ≤ −2.5 indicates high risk for osteoporosis, T-score < −1 and > −2.5 indicates moderate risk for osteoporosis, and T-score > −1.0 indicates low risk for osteoporosis [15,22].

Kolmogorov–Smirnov test was used to determine the normality of the data. Independent t-test determined the significant difference in basic characteristics of subjects. Receiver operating characteristic (ROC) curves were generated to assess the performance of QUS in identifying subjects with suboptimal bone health (osteopenia + osteoporosis) and osteoporosis using DXA as the reference. The suboptimal bone health group is important because individuals with osteoporosis and osteopenia are both susceptible to fragility fractures [35,36]. In the first ROC, subjects with osteopenia or osteoporosis were grouped as suboptimal bone health. In the second ROC, subjects were divided into the group with or without osteoporosis (which included normal and osteopenia). Sensitivity, specificity and AUC were determined. Correlation between DXA and QUS T-score (continuous data) was identified using Pearson’s correlation, while Cohen’s kappa statistics determined the agreement between the bone health status categorized by both techniques (ordinal data). A kappa value (κ) ≤ 0.20 indicates no agreement, while 0.21 < κ ≤ 0.39 indicates minimal agreement, 0.40 < κ ≤ 0.59 indicates weak agreement, 0.60 < κ ≤ 0.79 indicates moderate agreement, 0.80 < κ ≤ 0.90 indicates strong agreement and k > 0.90 indicates perfect agreement [37]. Optimal cut-offs of QUS T-score were obtained by tracing the coordinates of ROC curves and Youden’s Index (J = sensitivity + specificity - 1). The cut-offs with the highest Youden’s Index were selected as the optimal cut-offs [38]. Significance value was set at *p* < 0.05 (two-tailed). All statistical analyses were performed using Statistical Package for Social Science version 22.0 (IBM, Armonk, NY, USA).

## 3. Results

A total of 910 subjects were recruited in the study, but 20 of them were excluded for receiving thiazide diuretics, 32 for glucocorticoids, 4 for cancer treatment, 30 for hormone treatment, 12 for having mobility problems, 5 for hysterectomy before menopause and 21 for not completing study procedure. Finally, data from the remaining 786 subjects were included in the final analysis, of which 48.6% were men and 51.4% were women. The overall prevalence of osteoporosis determined by DXA was 12.3%, but it was higher in women compared to men. Based on QUS, more than half of the subjects were categorised as having risk for osteoporosis, followed by normal and high risk of osteoporosis (Table 1).

Kappa and correlation statistics revealed that the agreement and association for osteoporosis risk stratification between CM-200 and DXA were minimal for men, women and the overall study population. DXA T-scores correlated positively and significantly with QUS T-scores of the subjects. The strength of association between QUS and DXA T-scores was similar between women and men (Table 2).

The performance of CM-200 using DXA as reference was assessed through ROC analysis (Figure 1). At T-score < −1.0, CM-200 showed a sensitivity of 82.1% (95% CI: 77.9–85.7%), specificity of 51.5% (95% CI: 46.5–56.6%) and AUC of 0.668 (95% CI: 0.630–0.706; *p* < 0.001) in identifying subjects with suboptimal bone health (DXA T-score < −1). At T-score ≤ −2.5, CM-200 showed a sensitivity of 14.4% (95% CI: 8.1–23.0%), specificity of 96.1% (95% CI: 94.4–97.4%) and AUC of 0.553 (95% CI: 0.488–0.617; *p* = 0.093) in identifying subjects with osteoporosis (DXA T-score ≤ −2.5). Sub-analysis according to sex, age, ethnicity and BMI showed similar results, whereby the sensitivity and AUC of CM-200 in detecting subjects with suboptimal bone health were better than in detecting subjects with osteoporosis. The performance of CM-200 in determining men and women with suboptimal bone health or osteoporosis was found to be similar. For men, there was no obvious age trend for the performance of CM-200. However, the sensitivity of CM-200 was higher in older compared to younger women. Among men, the performance of CM-200 in detecting subjects with suboptimal bone health was better in Malay and Chinese men compared to Indians or other ethnic groups, whereas it did not differ among women. The performance of CM-200 also did not differ between the BMI groups in identifying subjects with suboptimal bone health (Table 3 and Table 4)

Modification of the QUS T-score of CM-200 was attempted to determine its optimal cut-off values in identifying subjects with suboptimal bone health or osteoporosis. At cut-off < −1.32, an improvement in the performance of CM-200 in identifying men with suboptimal bone health was observed compared to cut-off < −1. Similarly, at cut-off < −1.61, the performance of CM-200 in identifying men with osteoporosis also improved compared to cut-off ≤ −2.5. For women, the optimal QUS T-score cut-off value for suboptimal bone was < −1.37. At cut-off ≤ −1.43, the performance of CM-200 in identifying women with osteoporosis was improved compared to cut-off ≤ −2.5 (Table 5).

## 4. Discussion

The current study indicated that the agreement between QUS and DXA at the current cut-offs was significant but weak. Since limited studies reported the kappa coefficient between QUS and DXA, comparison with other devices is not possible. This study also found that QUS and DXA correlated significantly in predicting suboptimal bone health and osteoporosis among the subjects. At the current cut-offs, CM-200 could identify subjects with suboptimal bone health but not subjects with osteoporosis. The modified cut-offs improved the performance of CM-200 in identifying subjects with suboptimal bone health and osteoporosis. 

The performance of QUS (T-score < −1) in identifying subjects with suboptimal bone health (DXA T-score < −1) was fair based on AUC, sensitivity and specificity values (Table 3). Although being underweight, women and Chinese were reported to be predictors of osteoporosis and suboptimal bone health in the same cohort of subjects [27], and sub-analysis based on sex, age, ethnicity and BMI did not alter the results significantly (Table 3), except among Indian men and those from other ethnicities, probably due to small sample size (*n* = 41). On the other hand, QUS was not effective in identifying subjects with osteoporosis in the present study (Table 4). This observation is contradictory to earlier studies which reported a satisfactory performance of QUS (of different manufacturers) in identifying people with osteoporosis defined by T-score [21,39]. Among 109 Turkish men (mean age: 57.8 ± 13.7 years) and 131 women (mean age: 53.7 ± 11.9 years), both BUA and SOS indices of a calcaneal QUS device (Sahara, Hologic) demonstrated an acceptable performance (men: BUA = 0.661, SOS = 0.735; women: BUA = 0.712, SOS = 0.764) in identifying subjects with osteoporosis (DXA T-score ≤ −2.5) [40]. McLeod et al. (2015) reported good performance of a calcaneal QUS device (Lunar Achilles) in identifying patients with osteoporosis among 174 Canadian women (59.7 ± 6.7 years). The QUS device showed AUCs of 0.892 (using DXA T-score ≤ −2.5 at the femoral neck as reference) and 0.696 (using DXA T-score ≤ −2.5 at the lumbar spine as reference) at QUS T-score ≤ −2.5 [41]. The reasons for discrepancies between our studies and the aforementioned studies are not clear.

Since the use of WHO’s cut-offs based on DXA T-score for QUS assessment is not recommended [18], the present study has determined the optimal QUS T-score for CM-200 based on sex to identify subjects with suboptimal bone health and osteoporosis. The modified cut-offs of QUS T-score for CM-200 improved the sensitivity of QUS in identifying subjects at risk of osteoporosis and men with suboptimal bone health. Particularly, the performance of CM-200 in identifying subjects at risk of osteoporosis changed from ineffective to effective (Table 5). Some of the previous studies have applied modified QUS T-score to determine the optimal performance of QUS in identifying subjects at risk of osteoporosis [21,42,43,44]. Oral et al. (2019) found that at the modified QUS T-scores of −1.68 and −1.53 for men (mean age: 57.8 ± 13.7 years) and women (mean age: 53.7 ± 11.9 years), respectively, the sensitivity and specificity values obtained were around 70% [40]. Nevertheless, it should be emphasized that the comparison of the performance and cut-off values across QUS devices from different manufacturers is not recommended due to different algorithms or reference database used.

There are several limitations of this study to be addressed. A nonrandomized technique was used to recruit the subjects; thus generalization of the results of this study should be made with caution. However, the ethnic composition of the subjects resembles the ethnic demography in Klang Valley [30]. The subjects were healthier than the general population because subjects with strong secondary risk for osteoporosis had been excluded. Thus, the results of this study might be applied to individuals without strong secondary risk factors of osteoporosis only. The CM-200 device only generates SOS but does not generate other QUS indices, and its reference population is different from the DXA used in this study. Therefore, we cannot guarantee that the results of this study can be replicated in other populations with different characteristics or with QUS devices from other manufacturers. Subgroup analysis among subjects categorized as Indian and underweight may be underpowered due to the low sample size. The sample size was calculated on the assumption that the modified cut-offs for CM-200 could reach an AUC of 0.7. We recalculated the sample size needed for the group with the lowest AUC (0.685 for identification of women with osteoporosis; Table 5) and found that the sample size required was still within acceptable range (*n* = 128). Lastly, a validation cohort may be required to prove the effectiveness of the modified cut-off values generated in identifying individuals at risk of osteoporosis. Some of the aspects that should be investigated in future research include clinical significance and cost–benefit of using CM-200 in the screening of osteoporosis. Nevertheless, the information from this study can serve as a guide for future studies using CM-200 in public screening of bone health. 

## 5. Conclusion

The performance of the CM-200 calcaneal QUS device using cut-offs similar to DXA is fair in identifying subjects with suboptimal bone health, but it cannot identify subjects with osteoporosis. The performance of CM-200 improves significantly when a new set of cut-off values is adopted. Therefore, it is recommended that QUS devices should be optimized and their cut-off value should be validated before deploying them in the local setting to ensure optimal performance in identifying individuals at risk of osteoporosis.

## Figures and Tables

**Figure 1 diagnostics-10-00178-f001:**
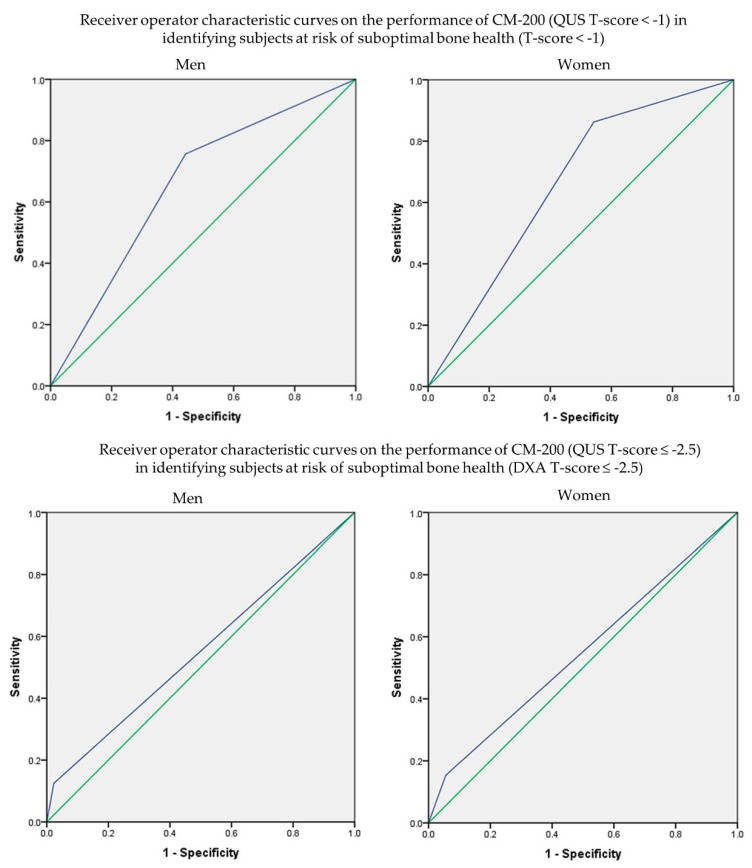
Receiver operating characteristic curves of QUS in identifying the risk of osteoporosis with reference to DXA.

**Table 1 diagnostics-10-00178-t001:** Basic characteristics of the subjects.

Variable of Interest	Mean (SD)		
	Men (*n* = 382)	Women (*n* = 404)	Overall (*n* = 786)
Age (years)	58.35 (9.41) ^a^	56.03 (8.70)	57.16 (9.12)
Weight (kg)	70.90 (10.78) ^a^	61.08 (12.30)	65.85 (12.56)
Height (m)	166.54 (9.67) ^a^	154.60 (5.50)	160.39 (9.84)
BMI (kg/m^2^)	25.42 (3.61)	25.54 (4.98) ^a^	25.48 (4.36)
**DXA T-score**			
Lumbar spine	0.05 (1.27)	−0.7 (1.40) ^a^	−0.4 (1.39)
Left hip	−0.05 (1.28)	−1.1 (1.29)	−0.8 (1.31)
**QUS T-score**			
Calcaneus	−1.05 (0.89)	−1.39 (0.83)	−1.2 (0.87)
	*n* (%)		
**Age Range (years)**			
40–50	87 (22.8)	115 (28.5)	202 (25.7)
51–60	127 (33.2)	156 (38.6)	282 (36.0)
61 and above	168 (44.0)	133 (32.9)	301 (38.3)
**Ethnicity**			
Malay	160 (41.9)	182 (45.0)	342 (43.5)
Chinese	181 (47.4)	182 (45.0)	363 (46.2)
Indians/Others	41 (10.7)	40 (9.9)	81 (10.3)
**Body Mass Index (kg/m^2^)**			
Underweight	173 (45.3)	177 (43.8)	65 (8.3)
Normal	26 (6.8)	39 (9.7)	350 (44.5)
Overweight	183 (47.9)	188 (46.5)	371 (47.2)
**Bone health status based on QUS**			
Low risk	164 (42.9)	108 (26.7)	272 (34.6)
Moderate risk	206 (53.9)	267 (66.1)	473 (60.2)
High risk	12 (3.1)	29 (7.2)	41 (5.2)
**Bone health status based on DXA**			
Normal	226 (59.2)	164 (40.6)	390 (49.6)
Osteopenia	124 (32.5)	175 (43.3)	299 (38.0)
Osteoporosis	32 (8.4)	65 (16.1)	97 (12.3)

^a^ Indicates a statistically significant difference (*p* < 0.05). BMI, body mass index; QUS, quantitative ultrasound; SD, standard deviation; WHO, World Health Organization.

**Table 2 diagnostics-10-00178-t002:** The agreement and correlation between QUS and DXA.

	κ	*p*-Value	r	*p*-Value
Men	0.222	< *0.001*	0.352	< *0.001*
Women	0.212	< *0.001*	0.372	< *0.001*
Overall	0.232	< *0.001*	0.384	< *0.001*

Significant *p*-values are italicized. κ, kappa coefficient; QUS, quantitative ultrasound; r, Pearson’s correlation coefficient.

**Table 3 diagnostics-10-00178-t003:** The performance of CM-200 in identifying subjects with suboptimal bone health (DXA T-score < −1).

QUS < −1 vs. T-score < −1														
Men								Women						
	Sen. (%)	95% CI (%)	Spe. (%)	95% CI (%)	AUC	95% CI	*p*-Value	Sen. (%)	95% CI (%)	Spe. (%)	95% CI (%)	AUC	95% CI	*p*-Value
Age														
40–50 years	78.6	59.1–91.7	67.8	54.4–79.4	0.732	0.619–0.845	*0.001*	81.6	65.7–92.3	53.2	41.5–64.7	0.674	0.573–0.776	*0.002*
51–60 years	57.1	41.0–72.3	50.6	39.5–61.6	0.539	0.432–0.645	0.479	82.5	73.4–89.5	44.1	31.2–57.6	0.633	0.540–0.725	*0.006*
61 years and above	83.7	74.2–90.8	52.4	41.1–63.6	0.681	0.599–0.763	< *0.001*	91.4	84.4–96.0	28.6	15.6–55.3	0.600	0.473–0.727	0.105
Ethnic group														
Malay	74.5	60.4–85.7	58.7	48.9–68.1	0.666	0.577–0.755	*0.001*	90.6	82.95–95.62	48.8	37.9–59.9	0.697	0.619–0.776	*< 0.001*
Chinese	78.9	69.4–86.6	52.3	41.3–63.2	0.656	0.576–0.737	*< 0.001*	83.9	76.19–89.86	37.9	25.5–51.6	0.609	0.518–0.700	*0.018*
Indians/Others	50.0	18.7–81.3	54.8	36.0–72.7	0.524	0.316–0.732	0.82	80.0	56.34–94.27	55.0	31.5–76.9	0.675	0.505–0.845	0.058
BMI														
Underweight and Normal	75.9	66.8–83.6	58.2	47.4–68.5	0.671	0.595–0.747	*< 0.001*	85.8	79.30–90.89	34.4	22.7–47.7	0.601	0.513–0.689	*0.021*
Overweight	75.0	60.4–86.4	54.1	45.3–62.7	0.645	0.557–0.734	*0.003*	87.1	78.0–93.36	52.4	42.4–62.4	0.697	0.622–0.773	*< 0.001*
Overall	75.6	68.1–82.2	55.8	49.0–62.3	0.657	0.602–0.712	*<0.001*	86.3	81.24–90.34	45.7	37.9–53.7	0.660	0.604–0.716	*< 0.001*
**Overall**														
	**Sen. (%)**	**95% CI (%)**		**Spe** **. (%)**	**95% CI (%)**		**AUC**	**95% CI (%)**		***p*-Value**				
	82.1	77.9–85.7		51.5	46.5–56.6		0.668	0.630–0.706		*< 0.001*				

Significant *p*-values are italicized. AUC, area under curve; CI, confidence interval; QUS, quantitative ultrasound; Sen., sensitivity; Spe., specificity

**Table 4 diagnostics-10-00178-t004:** The performance of CM-200 in identifying subjects with osteoporosis (DXA T-score ≤ −2.5)

QUS ≤ −2.5 vs. T-score ≤ −2.5														
Men								Women						
	Sen. (%)	95% CI (%)	Spe. (%)	95% CI (%)	AUC	95% CI	*p*-Value	Sen. (%)	95% CI (%)	Spe. (%)	95% CI (%)	AUC	95% CI	*p*-Value
Age														
40–50 years	0.0	0.0–84.2	95.3	88.4–98.7	0.476	0.087–0.865	0.910	0.0	0.0–52.2	97.3	92.2–99.4	0.486	0.233–0.740	0.918
51–60 years	28.6	3.7–71.0	100.0	97.0–100.0	0.643	0.393–0.893	0.205	15.0	3.2–37.9	95.6	90.6–98.4	0.553	0.409–0.696	0.445
61 years and above	8.7	1.1–28.0	97.2	93.1–99.2	0.53	0.398–0.662	0.648	17.5	7.3–32.8	89.2	81.1–94.7	0.534	0.425–0.643	0.538
Ethnic group														
Malay	11.1	0.3–48.3	98.0	94.3–99.6	0.546	0.338–0.753	0.646	20.0	6.8–40.7	92.4	87.0–96.0	0.562	0.433–0.691	0.322
Chinese	14.3	3.1–36.3	96.9	92.9–99.0	0.556	0.416–0.696	0.406	11.4	3.2–26.7	97.3	93.2–99.3	0.544	0.432–0.655	0.424
Indians/Others	0.0	0.0–84.2	100.0	91.0–100.0	0.5	0.085–0.915	1.000	20.0	0.5–71.6	91.4	76.9–98.2	0.557	0.269–0.845	0.683
BMI														
Underweight and Normal	11.5	2.5–30.2	97.7	94.2–99.4	0.546	0.421–0.672	0.449	17.6	8.4–30.9	94.5	89.9–97.5	0.561	0.466–0.656	0.188
Overweight	16.7	0.4–64.1	97.7	94.3–99.4	0.572	0.314–0.830	0.549	7.1	0.2–33.9	94.3	89.7–97.2	0.507	0.348–0.666	0.931
Overall	12.5	3.5–29.0	97.7	95.6–99.0	0.551	0.440–0.662	0.339	15.4	7.6–26.5	94.4	91.4–96.6	0.549	0.469–0.629	0.211
**Overall**														
	**Sen. (%)**	**95% CI (%)**		**Spe** **. (%)**	**95% CI (%)**		**AUC**	**95% CI**		***p*-Value**				
	14.4	8.1–23.0		96.1	94.4–97.4		0.553	0.488–0.617		0.093				

Significant *p*-values are italicized. AUC, area under curve; CI, confidence interval; QUS, quantitative ultrasound; Sen., sensitivity; Spe., specificity

**Table 5 diagnostics-10-00178-t005:** Modified cut-off values of QUS T-scores for CM-200 according to sex in determining subjects with suboptimal bone health (DXA T-score < −1) and osteoporosis (DXA T-score ≤ −2.5).

	DXA T-score < −1									DXA T-score ≤ −2.5								
	Modified QUS Cut-Off	Sen. (%)	95% CI (%)	Spe. (%)	95% CI (%)	Youden’s Index	AUC	95% CI	*p*-Value	Modified QUS Cut-Off	Sen. (%)	95% CI (%)	Spe. (%)	95% CI (%)	Youden’s Index	AUC	95% CI	*p*-Value
Men	< −1.32	61.5	53.4–69.2	77.4	71.4–82.7	0.389	0.695	0.640–0.750	*<0.001*	< −1.61	76.3	71.5–80.7	68.8	50.0–83.9	0.451	0.707	0.612–0.801	*<0.001*
Women	< −1.37	70.0	63.8–75.7	68.9	61.2–75.9	0.389	0.691	0.637–0.744	*<0.001*	< −1.43	83.1	71.7–91.2	54.0	48.5–59.4	0.371	0.685	0.620–0.750	*<0.001*
Overall	< −1.32	67.7	62.8–72.3	72.8	68.1–77.2	0.405	0.702	0.666–0.739	*<0.001*	< −1.42	79.4	70.0–86.9	61.8	58.1–65.5	0.412	0.706	0.654–0.758	*<0.001*

Notes: Cut-off values close to the sensitivity of 80% with reasonable specificity were chosen. Significant *p*-values are italicized. AUC, area under the curve; CI, confidence interval; QUS, quantitative ultrasound; Sen., sensitivity; Spe., specificity

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
