# Peer review of "The Performance of a Calcaneal Quantitative Ultrasound Device, CM-200, in Stratifying Osteoporosis Risk among Malaysian Population Aged 40 Years and Above"

_diagnostics, 2020, doi:10.3390/diagnostics10040178_

Round 1
Reviewer 1 Report
Review for Manuscript diagnostics-736731
General Comments: The design and presentation of this study is straightforward. The length is appropriate, nicely written, with very nice data presentation. While in a very specific age and lack of pre-existing disease population, the limitations are discussed. I have only minor comments below.
I have made specific comments below by line number
More Specific Comments:
Title – None
Abstract – None
Introduction
- Line 68 – Change “bone bones” to “bones”
- Line 70 – Change “anatomy” to “anatomical”
- Line 74-76 – Provide more information/results from the CM-200
Materials and Methods
- Line 119-120 – Expand on the “standard protocol” for DXA
Results – None
Discussion – None
Figures, Tables, and Legends – Very nice
Author Response
Dear Reviewer,
Thank you for reviewing our manuscript meticulously. We appreciated your constructive comments.
Dear Reviewer,
Thank you for reviewing our manuscript meticulously. We appreciated your constructive comments.
Response to comments of Reviewer 1
Manuscript title: The performance of a calcaneal quantitative ultrasound device, CM-200, in stratifying osteoporosis risk among Malaysian population aged 40 years and above.
Manuscript ID: diagnostics-736731
Thank you for reviewing our manuscript. We appreciated your comments, and our replies are listed below:
Comment 1: Line 68 – Change “bone bones” to “bones”
Thank you for your comment. The changes has been made as per suggested.
Comment 2: Line 70 – Change “anatomy” to “anatomical”
Thank you for your suggestion. The word has been changed as per requested.
Comment 3: Line 74-76 – Provide more information/results from the CM-200
Thank you for your comment. We have added some details about the device in Line 85-86.
Comment 4: Line 119-120 – Expand on the “standard protocol” for DXA
Thank you for your comment. We have elaborated on the protocol during DXA scan in Line 132-133.
Thank you for your time. We look forward to receiving your favourable reply.
Reviewer 2 Report
General comments:
- It is really needed to validate the performances of QUS?
- The performance metrics must be accompanied by associated 95% confidence interval. Also, more other metrics, especially with clinical significance, must accompany the results (doi:10.1155/2019/1891569).
Title:
- avoid using abbreviations in the title.
- it is so important the population and the age that need to be included in the title of the manuscript?
Abstract:
- Why new sets of cut-offs are needed?
- The number of patients is a result, not a material/method. Provide instead the inclusion and exclusion criteria and the delimitation in time for the study.
- Detection of an individual at risk is related to a screening test ... the proposed cut-off is with a high number of false positives meaning at least application of unnecessary tests or treatment.
- I do not agree with the conclusion; please rephrase for clarity and validity.
Introduction:
- "because 20‐40% of people suffering from hip fractures would die within a year and 10%" these values are correct for which population?
- The correct and used abbreviation is DEXA.
- Please provide the performances of DEXA.
- is the interpretation of the BMD the same in Asia as compared to WHO?
- "DXA is limited by its cost" which is the cost of the procedure?
- Which is the availability of DEXA in your country?
- define "relatively less costly"; please be specific.
- List the performances of QUS.
- "QUS remains controversial" please be specific.
- Was any QUS study conducted in your population? If yes, briefly present the reported results.
- Is the investigated population so special that must be part of the aim of the study?
- End this section with the aim of the study. Move the last sentence in this section to Discussion.
Materials and Methods:
- The number of patients is a result.
- Which was the source of negative / positive cases = 689/97?
- It is well known that US is operator dependent. What was done to assure a small variability between examiners?
- The staff performing QUS was aware of the DEXA result?
- It is not clear if the QUS and DEXA are quantitative continuous values or ordinal.
- "osteopenia or osteoporosis were grouped as suboptimal" why?
- Pearson correlation coefficient is not proper to be used to analyze the results (see doi:10.1155/2019/1891569)
- The interpretation of kappa statistics must be provided.
- Which was the method applied to identify the treshold ?
Results:
- Report the subjects included in the analysis of the available population.
- significantly higher?
- Table 1: Splitting the sample is Men and Women must have a rationale. It is expected to find differences? If yes, comparisons between men and women must be included in Table 1.
- Table 1: I see no reason to provide the results of weight and height.
- The values of Kappa statistics are very low but statistically significant differences by zero due to the presence of a large sample size.
- The results must be reported according to those that support the analysis of performances of a diagnostic test.
- To allow a proper interpretation, the performance metrics must be reported with the associated 95% confidence intervals.
- Table 3 and 4 and 5: do not include the units of measurements in the body of the table.
- The p-value must be adjusted when subgroups analysis is performed.
- I do not agree with you that a Sp of 51 is reasonable.
- Do not discuss your results in this section.
Discussions:
- Start this section with the main finding.
- Do not duplicate information in this section (the written information was already provided in previous sections).
- Discuss your results with reference to your tables and figures in the light of the scientific literature.
- The effects of underweight on bone must be discussed.
- Discuss the effect of underweight on the bone structure.
- The generalizability of the results to other population and/or other QUS devices than the one used in this study must be properly discussed.
Author Response
Dear Reviewer,
Thank you for reviewing our manuscript meticulously. We appreciated your constructive comments.
Response to comments of Reviewer 2
Manuscript title: The performance of a calcaneal quantitative ultrasound device, CM-200, in stratifying osteoporosis risk among Malaysian population aged 40 years and above.
Manuscript ID: diagnostics-736731
Thank you for reviewing our manuscript. We appreciated your comments, and our replies are listed below:
Title and Abstract
Comment 1: Is it really needed to validate the performances of QUS? The performance metrics must be accompanied by associated 95% confidence interval. Also, more other metrics, especially with clinical significance, must accompany the results (doi:10.1155/2019/1891569).
Thank you for your comments.
- This study is necessary because QUS has been used widely in the screening of osteoporosis. The instrument of interest in this study, CM-200 have been used in medical literature yet its performance against DXA has yet to be reported. Since the performance of QUS varies according to device and population investigated, we think the performance of CM-200 should be validated.
- We have added the 95% confidence interval for the AUC as per your suggestion.
- Thank you for the suggested read (doi:10.1155/2019/1891569). We acknowledge that the study is not the most complete. The other metrics, including clinical significance and cost benefits, should be investigated in further studies. We have included these issues in the limitation paragraph.
Comment 2 : Avoid using abbreviations in the title. Is it so important the population and the age that need to be included in the title of the manuscript?
Thank you for your comment. CM-200 is the name of QUS device examined in the study, not an abbreviation. It is important to include the population and age group in the title because the performance may vary in other populations.
Comment 3 : Why new sets of cut-offs are needed?
Thank you for your comment. The new set of cut-offs are needed to optimize the performance of CM-200 in identifying individuals with increased risk of osteoporosis and accurate classification of the bone health status of subjects. The current cut-offs using World Health Organization (WHO) guidelines are based on DXA, which should not be implemented in QUS machines (doi: 10.1016/j.jocd.2007.12.011).
Comment 4: The number of patients is a result, not a material/method. Provide instead the inclusion and exclusion criteria and the delimitation in time for the study.
Thank you for your comment. We have moved the number of patients to the Results section. For your information, the inclusion and exclusion criteria are mentioned in Line 98-103, but they are not included in the abstract due to the word limit.
Comment 5: Detection of an individual at risk is related to a screening test ... the proposed cut-off is with a high number of false positives meaning at least application of unnecessary tests or treatment.
Thank you for your comment. For your information, CM-200 is intended to be used as a screening tool for osteoporosis. The subjects tested with CM-200 will be subjected to DXA to confirm their bone health status, rather than initiating treatment immediately. As much as we wish to achieve both high sensitivity and specificity in a tool, these two indices are usually inversely proportional to each other in reality. A tool with high sensitivity usually faces the problem of low sensitivity. The cut-offs selected for CM-200 in this study enable it to achieve a reasonable level of sensitivity and specificity. Since the purpose of a screening tool is to identify high-risk individuals, its sensitivity should be high.
Comment 6: I do not agree with the conclusion; please rephrase for clarity and validity.
We asked the reviewer to kindly reconsider this comment. The conclusion is divided into two parts:
“The modified cut-off values significantly improved the performance of CM-200 in identifying individuals with osteoporosis”
We show that with the modified cut-offs, the performance of QUS improves from not effective in detecting individual with osteoporosis (sensitivity 14.4%; specificity 96.1%; AUC 0.553 (95% CI: 0.488-0.617); p>0.05) to effective (sensitivity 80.4%; specificity 61.1%; AUC 0.705 (95% CI: 0.654-0.757); p<0.001). Thus, this part of the conclusion is supported by the data.
“Since these values are device-specific, optimisation is necessary for accurate detection of individuals at risk for osteoporosis using QUS”
This recommendation to alert the reader of the potential limitations of the new cut-offs designated for CM-200.
Introduction
Comment 7: “because 20‐40% of people suffering from hip fractures would die within a year and 10%" these values are correct for which population?
Thank you for your comment. It is a pooled percentage of hip fractures among population from the United States (doi: 10.1093/oxfordjournals.aje.a116756), Australia (doi:10.1016/S0140-6736(98)09075-8), Asian countries and Middle East Crescent (doi:10.1007/s00198-004-1627-0). We have added the information on the regions referred to in the text.
“A pooled analysis of data from the United States, Australia, Asia and the Middle East Crescent shows that hip fractures are particularly devastating because 20-40% of people suffering from hip fractures would die within a year”
Comment 8: The correct and used abbreviation is DEXA
Thank you for your comment. DEXA is formerly used as the abbreviation for dual-energy X-ray absorptiometry until the International Society for Clinical Densitometry (ISCD) proposed the abbreviation DXA in 2003 (https://www.iaea.org/resources/rpop/health-professionals/other-specialities-and-imaging-modalities/dxa-bone-mineral-densitometry).
Hence, we would like to retain the use of DXA as the abbreviation for dual-energy X-ray absorptiometry in the manuscript.
Comment 9: Please provide the performances of DEXA.
We apologise for not understanding this particular comment. DXA is the gold standard of measuring bone mineral density and diagnosis of osteoporosis, so we are not sure which performance the reviewer is referring to. Nevertheless, we cited a few studies on the performance of DXA in predicting fracture (Line 66) and monitoring treatment.
Comment 10: Is the interpretation of the BMD the same in Asia as compared to WHO?
Thank you for your comment. T-score is used to classify the individual bone health status rather than absolute BMD. The interpretation of T-score is the same of Asian population because the reference used for calculation is the normal population of the same gender and ethnicity (doi: https://doi.org/10.1093/qjmed/hcn022).
Comment 11: "DXA is limited by its cost" which is the cost of the procedure?
Thank you for your comment. The cost refers to DXA machine acquisition and procedure. This has been added in Line 68.
Comment 12: Which is the availability of DEXA in your country?
Thank you for your comment. There are 100 DXA scanners available in Malaysia up to 2012 (Asia-Pacific Regional Audit 2013) and this statement has been added in the manuscript (Line 69). A partial list of the DXA machines available in Malaysia can be found through this link: http://www.osteoporosis.my/resources/hospital.asp.
Comment 13: define "relatively less costly"; please be specific
Thank you for your comment. We have added “relatively less costly methods of bone health screening compared to DXA”.
Comment 14: List the performances of QUS.
Thank you for your comment. The performance of QUS has been mentioned in Line 76-77, whereby we indicate that its indices associated significantly with bone mineral density, microarchitecture and fracture risk.
Comment 15: "QUS remains controversial" please be specific.
Thank you for your comment. We have explained the controversies in Line 81-83.
“the use of WHO’s T-score cut-offs for osteoporosis and osteopenia in QUS remains controversial because the cut-off values of < -1 and ≤ -2.5 are established based on DXA and the skeletal properties examined are different between QUS and DXA”
Comment 16: Was any QUS study conducted in your population? If yes, briefly present the reported results.
Thank you for your comment. For your information, this is the first report of the QUS bone indices of the study population. We have reported the QUS values in Table 1.
Comment 17: Is the investigated population so special that must be part of the aim of the study?
Thank you for your comment. We include the details of the populations as part of the aim of the study to inform the readers about the generalizability of the results of the current study.
Comment 18: End this section with the aim of the study. Move the last sentence in this section to Discussion.
Thank you for your suggestion. We have removed the last sentence in the introduction to the last line of discussion.
Materials and Methods
Comment 19: The number of patients is a result.
Thank you for your comment. We have deleted the number of subjects in the Materials and Methods section and insert it in the Results section.
Comment 20: Which was the source of negative / positive cases = 689/97?
Thank you for your comment. Since this study is a secondary analysis of the main study (doi: https://doi.org/10.3390/ijerph16101787; https://doi.org/10.3390/ijerph17020384), we have already known the number of subjects with normal bone health/osteopenia (negative case=689) and with osteoporosis (positive case) determined by DXA.
Comment 21: It is well known that QUS is operator dependent. What was done to assure a small variability between examiners?
Thank you for your comment. For your information, only one trained technician operated the QUS machine throughout the study.
Comment 22: The staff performing QUS was aware of the DEXA result?
Thank you for your comment. The QUS and DXA machines were handled by different technicians and they are not aware of the results of the subject from another machine.
Comment 23: It is not clear if the QUS and DEXA are quantitative continuous values or ordinal.
The T-scores of QUS and DXA are continuous data and used for correlation analysis. The bone health status according to QUS and DXA results are ordinal data and used for kappa statistics.
Comment 24: "osteopenia or osteoporosis were grouped as suboptimal" Why?
Thank you for the comment. We group “osteopenia” and “osteoporosis” group together as “suboptimal bone health” because fragility fractures can occur in individuals with osteopenia and osteoporosis. Both of the groups should undergo DXA scan to confirm their bone health status.
Comment 25: Pearson correlation coefficient is not proper to be used to analyze the results (see doi:10.1155/2019/1891569)
Thank you for your concern. Pearson correlation is used as a preliminary measure to study the association between QUS T-score (continuous data) and DXA T-score (continuous data). It has been used by other studies to study the relationship between DXA and QUS indices (doi: 10.1007/s00774-013-0474-5;10.1371/journal.pone.0145879)
Comment 26: The interpretation of kappa statistics must be provided.
Thank you for your suggestion. The interpretation of kappa statistic has been added in Line 152-156.
Comment 27: Which was the method applied to identify the threshold?
Thank you for your comment. The method of identifying the threshold has been included in the manuscript.
Line 155-156 : “Optimal cut-offs of QUS T-score were obtained by tracing the coordinates of ROC curves which are closest to a sensitivity value of 80% and a reasonable specificity value.”
Results
Comment 28: Report the subjects included in the analysis of the available population. significantly higher?
Thank you for your comment. The total number of subjects recruited and included in the final analysis has been added in Line 157-161.
Comment 29: Splitting the sample is Men and Women must have a rationale. It is expected to find differences? If yes, comparisons between men and women must be included in Table 1.
Thank you for your comment. The results are split based on sex because other studies have reported that the modified QUS cut-off values are different between men and women (doi: 10.3346/jkms.2009.24.2.232; https://academic.oup.com/jcem/article/97/3/800/2536329). We have considered your advice and tabulated the characteristics of the subjects based on sex in Table 1.
Comment 30: I see no reason to provide the results of weight and height.
Thank you for the comment. Weight/height and BMI are potential factors influencing bone health of the subjects, so we prefer the data to be shown to the readers.
Comment 31: The values of Kappa statistics are very low but statistically significant differences by zero due to the presence of a large sample size.
Thank you for the comment. We knowledge that the agreement between QUS and DXA at the existing cut-offs is minimal based on the Kappa statistics. This warrant modification of the QUS cut-offs to achieve better agreement with DXA results.
Comment 32: The results must be reported according to those that support the analysis of performances of a diagnostic test.
Thank you for the comment. We have reported the sensitivity, specificity and AUC with 95% confidence interval in the text and tables. This has been done in other studies.
Comment 33: To allow a proper interpretation, the performance metrics must be reported with the associated 95% confidence intervals.
Thank you for the comment. We have reported the AUC values with 95% confidence interval in the text and tables. This has been done in other studies (doi: https://academic.oup.com/jcem/article/97/3/800/2536329 - see Table 5; 10.2147/TCRM.S145519 – see Table 4-5).
Comment 34: Table 3 and 4 and 5: do not include the units of measurements in the body of the table.
Thank you for your comment. We have removed the unit as per requested.
Comment 35: The p-value must be adjusted when subgroups analysis is performed.
Thank you for your reminder. We understand the concerns of the reviewer because statistical power within the subgroup may be reduced. Our sample size calculation shows that we need 155 subjects (19+135) for the analysis. Referring to Table 1, the number of subjects recruited should be sufficient for subgroup based on sex, age, ethnic (except Indian) and BMI (except underweight) groups. We have added in the limitation paragraph that subgroup of Indian and underweight might be underpowered.
Comment 36: I do not agree with you that a Sp of 51 is reasonable.
Thank you for your concern. The specificity of 51.5% is the best we could get at the sensitivity of 82.1%. Further increasing the specificity will decrease the sensitivity, which defeats the purpose of a screening tool.
Comment 37: Do not discuss your results in this section.
Thank you for your comment. We did not discuss the results in the Results section.
Discussions
Comment 38: Start this section with the main finding.
Thank you for your comment. We deleted the first few lines of the discussion which contains information from the introduction. This section now begins with a discussion on the overall correlation and agreement between QUS and DXA to give a general idea on the concordance between the two techniques.
Comment 39: Do not duplicate information in this section (the written information was already provided in previous sections).
Thank you for your comment. We deleted the first few lines of the discussion which contains information from the introduction. We have also removed the numerical values of the results from this section.
Comment 40: Discuss your results with reference to your tables and figures in the light of the scientific literature.
Thank you for your comment. In line with the changes above, we added the tables to refer to when referring to the results of this study.
Comment 41: The effects of underweight on bone must be discussed.
Comment 42: Discuss the effect of underweight on the bone structure.
With response to comment 41 and 42, the effects of underweight on bone health has been reported and discussed in our previous publication (doi: 10.3390/ijerph17020384). In this study, we have demonstrated that BMI categories did not influence the performance of QUS, so we did not repeat the same discussion on bone mass/structure. Nevertheless, we added that:
Line 230-231:“Although being underweight, women and Chinese were reported to be predictors of osteoporosis and suboptimal bone health in the same cohort of subjects, sub-analysis basis based on sex, age, ethics and BMI did not alter the results significantly”
Comment 43: The generalizability of the results to other population and/or other QUS devices than the one used in this study must be properly discussed.
Thank you for your concern. We have addressed the issues of generalizability of the results of this study in the limitation paragraph. Mainly:
- The study population is an urban population in central Malaysia consisting of three major ethnic groups.
- The subjects were free from strong risk factors of osteoporosis.
- The QUS machines from different manufacturers used different technologies and algorithms, the cut-offs of CM-200 may not be applicable to QUS machines from other manufacturers.
We have advised the readers to take into account these issues in interpreting the results of this study.
Thank you for your time. We look forward to receiving your favourable reply.
Round 2
Reviewer 2 Report
The authors improved their manuscript but some of their answers were not included in the revised manuscript:
- To allow the reader top proper interpret the Se and Sp please provide the 95% confidence intervals for these metrics along the manuscript.
- In sample size calculation, please put the reference for negative / positive cases
- please clearly specify that there was only one trained technician.
- Clearly specify that the two trained technicians were blinded to the report of previous analysis.
- Provide in the revised manuscript the reason for the inclusion of osteopenia or osteoporosisin the same group.
- instead of saying "reasonable specificity" provide the desired value or the range of the desired values.
- there are still duplicated results in text and tables (e.g. BMI, % for ethnicity, etc.).
- Table 1: a space before the "(" is missing.
- Generally, the Youden index is used to identify the best cut-off.
- Your target area under the curve (AUC) = 0.7 (for this was the sample size calculated) but the reported values are less than 0.7. This need to be discussed in the Discussion section.
- Do not start the Discussion sentence with the summary or rational of your study; provide here the main result of your study.
Author Response
Dear reviewer,
Thank you for your constructive comments. We have considered and addressed each of them in the attached response sheet.
Thank you.
Title: The performance of a calcaneal quantitative ultrasound device, CM-200, in stratifying osteoporosis risk among Malaysian population aged 40 years and above
ID: diagnostics-736731
Thank you for reviewing our manuscript. We appreciate your constructive comments and they are responded as the following.
|
Comment |
Response |
|
To allow the reader top proper interpret the Se and Sp please provide the 95% confidence intervals for these metrics along the manuscript. |
Thank you for your suggestion. We have added the 95% CI values for sensitivity and specificity in the tables and text. |
|
In sample size calculation, please put the reference for negative / positive cases. |
Thank you for your comment. We have added the reference [17] in sample size calculation. |
|
Please clearly specify that there was only one trained technician. Clearly specify that the two trained technicians were blinded to the report of previous analysis. |
Thank you for your comment. We have added the following: Line 124-125: “DXA… operated by a single trained technician, blinded to the results of QUS, throughout the study period” Line 136-137: “CM-200… operated by another trained technician, blinded to the results of DXA, throughout the study period” |
|
Provide in the revised manuscript the reason for the inclusion of osteopenia or osteoporosis in the same group. |
Thank you for your comment. We have added the following: Line 147-148: “The suboptimal bone health group is important because individuals with osteoporosis and osteopenia are both susceptible to fragility fractures [35,36].” |
|
Instead of saying "reasonable specificity" provide the desired value or the range of the desired values. |
Thank you for your suggestion. We have removed the phrase “reasonable specificity” because we have adopted your suggestion to use Youden’s index as a criterion to select the best cut-off values. |
|
There are still duplicated results in text and tables (e.g. BMI, % for ethnicity, etc.). |
Thank you for your reminder. We have deleted all redundant numerical results from the text. |
|
Table 1: a space before the "(" is missing. |
Thank you for your reminder. We have added a space before () in Table 1. |
|
Generally, the Youden index is used to identify the best cut-off. |
Thank you for your suggestion. We have adopted Youden’s index as a criterion to select the best cut-off. The changes are reflected in the Results and Table 5. |
|
Your target area under the curve (AUC) = 0.7 (for this was the sample size calculated) but the reported values are less than 0.7. This need to be discussed in the Discussion section. |
Thank you for your comment. We added the following: Line 262-265: “The sample size was calculated on the assumption that the modified cut-offs for CM-200 could reach an AUC of 0.7. We recalculated the sample size needed for the group with the lowest AUC (0.685 for identification of women with osteoporosis; Table 5) and found that the sample size required was still within the acceptable range (n = 128).” |
|
Do not start the Discussion sentence with the summary or rational of your study; provide here the main result of your study |
Thank you for you comment. We have deleted the rationale of the study. The first paragraph now only contains a summary of results. |
We thank you again for your meticulous review.
Round 3
Reviewer 2 Report
NA